# Promoting User Participation of Shared Mobility in the Sharing Economy: Evidence from Chinese Bike Sharing Services

**Liguo Lou [1], Lin Li [2], Sung-Byung Yang [2] and Joon Koh [3,\*]**

1 College of Economics and Management, Ningbo University of Technology, 201 Fenghua Road, Jiangbei District, Ningbo, Zhejiang 315211, China; alexlou87@hotmail.com
2 School of Management, Kyung Hee University, 26 Kyungheedae-Ro, Dongdaemun-Gu, Seoul 02447, Korea; lilin@khu.ac.kr (L.L.); sbyang@khu.ac.kr (S.-B.Y.)
3 School of Business Administration, Chonnam National University, 77 Yongbong-Ro, Buk-Gu, Gwangju 61186, Korea
\* Correspondence: kjoon@chonnam.ac.kr; Tel.: +82-62-530-1459

**Abstract:** User participation plays a critical role in the business success of shared mobility services. This study classifies user participation behavior into two different types (in- and extra-role participations), integrates the motivation–opportunity–ability (MOA) model and social exchange theory (SET) to identify key antecedents, and empirically examines the influences of user–user, user–provider, and user–service interaction-related factors on user participation in the context of bike sharing services. The results of structural equation model analysis with 438 bike sharing service users in China reveal that altruism, rewards, and user knowledge enhance both in- and extra-role participations, whereas perceived ease of use promotes only user in-role participation, and both user satisfaction and commitment increase only user extra-role participation. Rewards are also found to promote user satisfaction, ultimately increasing user commitment. This study contributes to the body of knowledge on value co-creation and customer cooperation behavior in the sharing economy and provides practical implications to both managers of bike sharing services and policymakers for urban transportation and ICT-enabled sustainable development.

**Keywords:** sharing economy; shared mobility; bike sharing; value co-creation; user participation; motivation–opportunity–ability model; social exchange theory





## 1. Introduction

With the COVID-19 pandemic, platform- or access-based business models have dramatically changed people's lives, accelerating the sharing economy. A number of new information and communications technologies (ICTs) have facilitated service innovation that has contributed to the explosive growth of the sharing economy [1,2]. The sharing economy is defined as a scalable socio-economic system based on technology-enabled platforms that provide users with temporary access to tangible and intangible resources that can be crowdsourced [3]. These emerging business models consist of numerous typologies that challenge conventional ownership-based businesses, ushering in changes to various aspects of people's everyday lives such as transportation, financial and banking, accommodation, tourism, and entertainment industries.

Shared mobility is a representative example of the sharing economy, which refers to the shared use of bikes, motorcycles, cars, or other transportation vehicles, contributing to a number of social, environmental, and transportation cost-related benefits [4]. Among these shared mobility services, the commercial bike sharing service has grown to become a sustainable transportation alternative for millions across the globe, providing a low-carbon solution to the "last mile" problem [5–7]. As Shaheen et al. [4] suggested, bike sharing services primarily comprise of three models: station-based, dock-less, and hybrid models. In a station-based model, users are required to return bikes to the bike stations, while in

a dock-less model, users can pick up and return them within a predefined geographic region. A hybrid model integrates the characteristics of both models. In China, most bike sharing services have been adopting a mobile technology-based dock-less model that enables users to pick up and return bikes anywhere within an operating zone via mobile applications. A coin has two sides and so does the dock-less model; this free-floating bike sharing system enhances user flexibility and convenience, but at the same time, it gives rise to high operating costs with respect to re-allocating bikes for users on demand, monitoring users' misbehaviors, fixing bikes, and so forth. Hellobike is a firm that survives in a fierce competition in China [8]. In order to improve its business performance, Hellobike has implemented a number of user-centered marketing activities, such as providing membership cards to users, employing customer relationship management (CRM) programs, and encouraging users to report bike defects and/or other users' violations. Moreover, big data analytics has made it possible to clearly inform users of proper parking zones for pick-ups and returns and minimize operating costs by allowing users to do the tasks employees have to do.

Bike sharing, as one representative type of shared mobility, is essentially an access-based service that has transformed the understanding of possession/ownership and has reminded the importance of individual obligations that underlie shared objectives. Access-based services refer to services that allow users to access a resource for a defined period of time in return for an access payment while the service provider retains full legal ownership of the resource [9,10]. According to Belk [2] and Hartl et al. [11], the success of an access-based service relies largely on customer (user) cooperation/participation, which may provide users with a sense of responsibility, and thus loyalty, to a particular service. This reliance, however, is not without risk; the efficiency of this business model can be damaged by negative and opportunistic user behaviors, such as engaging in moral hazards and damaging hired items [11,12]. The impact of these behaviors can even extend beyond the confines of a particular business, affecting society by causing a waste of resources and a weakening of social order. Practitioners and academics have thus taken on the task of exploring various approaches to encourage customer (user) cooperation and participation.

Access-based services are technology-enabled self-services without employees' supervision in which the users play the role of "partial employees" and actively participate in the service delivery process [9]. According to Auh et al. [13], customer participation can occur when a customer is interacting with the service provider or even when the customer is alone. Previous studies have defined customer participation as voluntary cooperation behavior through which customers play the proactive role of partial employees in which the co-created service or product is both transferred to and owned by customers (e.g., [13,14]). However, in the context of access-based services, focusing only on user's voluntary behavior is insufficient as a service that can only be accessed by users is likely to lead to a lack of psychological ownership, triggering user misbehavior [9]. Therefore, to make this service operate successfully, users' required behavior is necessary in addition to their voluntary behavior. As Organ [15] suggested that employee behavior in an organization can be divided into discretionary (voluntary) and required behaviors, this study attempts to adopt this distinction to provide a more fine-grained account of user participation in a shared mobility environment.

In an attempt to contribute to the exploration of user participation in shared mobility services, this study classifies user participation into two different types (i.e., user in-role and extra-role participations) based on the discussion of value co-creation and customer citizenship behavior (CCB). In addition, drawing upon the motivation–opportunity–ability (MOA) model and social exchange theory (SET), this study aims to identify several key antecedent factors (i.e., altruism, rewards, perceived ease of use, user knowledge, user satisfaction, and user commitment) and validate the specific and different mechanisms to enhance user participation in the context of shared mobility services. Continuing on the path established by [3,10,16] in the specific context of access-based services, this study focuses on user–user, user–provider, and user–service interactions to identify key factors

that have the potential to enhance two different types of user participation. To empirically examine these relationships, this study focuses on the commercial bike sharing services in China as this industry has seen a rise, fall, and resurgence in the span of only a few years, during which the two types of user participation play critical roles in determining business performance.

The rest of the paper is organized as follows. Section 2 presents conceptual background and hypotheses development corresponding to the research purposes, respectively. The research method is explained in Section 3, while the results of data analyses and hypotheses tests are presented in Section 4. Lastly, the study findings, implications for both theory and practice, and limitations with future research directions are discussed in Section 5.

## 2. Conceptual Background and Hypotheses Development

### 2.1. User Participation

Businesses that can convince customers to actively collaborate with them to create value clearly hold a competitive advantage [17,18]. A service-dominant (S-D) logic suggests that the customer is always a co-creator of value, implying that customer collaboration is a key driver for firms that wish to provide more value and improve their business performance [19,20]. Value co-creation is the main premise of customer participation, which is defined as the extent to which customers are involved in producing and delivering a service by following a set of prescribed actions [14]. The concept of customer participation is distinct from other similar constructs (i.e., co-production, co-creation, and customer engagement) as customer participation is considered voluntary behavior that occurs without employee supervision [13].

On the other hand, as customers actively engage in the service delivery process and play the role of "partial employee," many researchers have turned their attention to CCB (e.g., [21,22]), which is drawn from Organ's [15] theory of organizational citizenship behavior (OCB). OCB refers to employee discretionary behavior that is helpful but not required by organizations, as opposed behavior required for the performance of in-role tasks [15,23]. Adopting and adapting this distinction, Groth [21] proposed the theory of CCB, which explores the role of customer cooperation in the act of co-production.

Using the lens of both value co-creation and CCB can aid in the exploration of customer cooperation behavior. Table 1 summarizes previous studies that have applied value co-creation and/or CCB perspectives to address relevant issues. Yi and Gong [24] developed a customer value co-creation scale, suggesting that the process of co-creation includes customer in-role participation and CCBs. Dennis et al. [25] adopted Yi and Gong's [24] two dimensions of value co-creation to examine their mediating roles between customer social exclusion and customer well-being in the context of shopping via multiple channels. Bartikowski and Walsh [26], Anaza [27], and Balaji [28] applied this CCB theory to the field of relationship marketing and examined various aspects of the antecedents of CCB. Yi and Gong [29] argued that enhancing CCBs and decreasing customer dysfunctional behaviors are clearly both important for firms. In particular, Yen et al. [22] combined value co-creation and CCB to conceptualize customer participation in terms of in- and extra-role participations in the context of virtual communities.

This study adopts the concepts of user in- and extra-role participations to examine user cooperation behavior in the context of commercial bike sharing services. User *in-role participation* refers to user behavior that is required and expected for the successful completion of a bike sharing service transaction, whereas *extra-role participation* describes user voluntary behavior that is not necessarily required but is critical in the maintenance and development of the bike sharing service. Enhancing user in- and extra-role participations can contribute to providing solutions that can decrease user misbehavior and promote value co-creation. This study views user in- and extra-role participations as having equal importance and aims to identify antecedent factors that can enhance them.

**Table 1.** Previous research on value co-creation and customer cooperation behavior.

| Authors | Research Focus | Customer Cooperation Behavior | |
| --- | --- | --- | --- |
| | | **Required, Expected** | **Voluntary, Discretionary** |
| Groth [21] | Comparing the different impacts of customer satisfaction and socialization on customer co-production and CCB | Required customer co-production behavior | CCBs (Recommendation, Helping, Providing feedback) |
| Yi and Gong [29] | Investigating the effects of customer justice perceptions on CCB and customer dysfunctional behavior | Opposite to required behavior: customer dysfunctional behavior | CCB |
| Bartikowski and Walsh [26] | Examining the impacts of corporate reputation as well as customer loyalty on CCB | - | CCBs (Helping other customers, Helping the company) |
| Yen et al. [22] | Conceptualizing and classifying customer participation and investigating their predictors in the context of online consumption communities | In-role participation | Extra-role participation (Recommendation, Helping others, Providing feedback) |
| Yi and Gong [24] | Customer value co-creation behavior scale development and validation | In-role participation (Information seeking, Information sharing, Responsible behavior, Personal interaction) | CCBs (Feedback, Advocacy, Helping others, Tolerance) |
| Anaza [27] | Examining the impact of customer personality on interpersonal relationships with service providers and CCB | - | CCBs (Recommendation, Helping others, Firm facilitation) |
| Balaji [28] | Examining the effects of relationship value, quality, and strength on CCB | - | CCB |
| Dennis et al. [25] | Examining the value co-creation mechanism through multiple shopping channels | In-role participation | CCB |
| *This Study* | *Conceptualizing user cooperation behaviors with user participation, and providing empirical evidence of what factors enhance user participation in the context of commercial bike sharing* | *User cooperation can be identified as user in-role and extra-role participations, which aims to provide appropriate approaches to decrease user misbehavior and increase business performance.* | |

### 2.2. User Participation in Shared Mobility

Eckhardt et al. [3] proposed five primary characteristics of the sharing economy: temporary access, transfer of economic value, platform mediation, expanded consumer role, and crowdsourced supply. Shared mobility (e.g., bike sharing) shares almost all of these characteristics, which is one of the most representative types of the sharing economy [3,30]. Compared with the ownership-based consumption, the shared mobility consumption is characterized as "what's mine is yours," implying that the firm's business performance and user experience quality heavily depend on user cooperative behaviors such as user in- and extra-role participations.

Previous research has noted that customer participation occurs when customers engage in interactions with other customers, service providers, and/or products/services themselves (e.g., [13,14,22]). This study, following Schaefers et al. [10], Park and Armstrong [16], and Eckhardt et al. [3], proposes that user participation is influenced by user–user, user–provider, and user–service interactions. First, regarding the user–user interaction, this study suggests *altruism* as the only but strong antecedent of user participation in the context of bike sharing services. Although previous studies have argued that a

commercial sharing program itself is not altruistic and is directed by negative reciprocity norms between the user and the firm (service provider) [1,9], this study holds that altruistic considerations among users become more apparent and enable users to create mutual value as a result of the permanency of altruism, a universal human characteristic. Second, regarding the user–provider interaction, as customer participation is a reciprocal outcome of a firm's value co-creation strategy and management of customer relationships [20,31], this study suggests that offering *rewards* in order to strengthen relationships with users could enhance user participation. In addition, drawing on SET, this study suggests *user satisfaction* (as a "cold" dimension of relationship quality) and *user commitment* (as a "hot" dimension of relationship quality) as other antecedents formed through user–provider interactions. Lastly, regarding the user–service interaction, as bike sharing services are technology-enabled self-services without employee supervision, this study follows Balaji and Roy [32] to propose that *perceived ease of use* can be a determinant of user participation. Moreover, following the S-D logic of Vargo and Lusch [20], *user knowledge* about how to use the service appropriately and comply with the relevant rules can be another determinant of user participation.

Based on these discussions, this study integrates the MOA model and SET to validate the importance of different factors of three types of interactions (i.e., user–user, user–provider, and user–service interactions) in enhancing two different types of user participation (i.e., user in-role and extra-role participations). The hypotheses regarding these relationships are developed in the following two subsections.

### 2.3. Antecedents from the MOA Model

MacInnis and Jaworski [33] and MacInnis et al. [34] proposed the MOA model to stress that a consumer's level of processing an advertisement is affected by her/his motivation, opportunity, and ability to process brand information. In their research, motivation refers to a consumer's desire or readiness to perform actions, opportunity means situational factors that can be impeditive or conducive to achieving a goal, and ability indicates a consumer's skills or proficiencies in conducting a behavior. The MOA model indicates that a consumer's behavior can be influenced by a firm's marketing strategies and activities. Researchers have utilized their work by applying this model in explaining customer engagement in value co-creation (e.g., [35,36]).

As discussed above, it is suggested that user motivation that includes altruism and expected rewards is likely to enhance user participation. Altruism is other-oriented intrinsic motivation that aims to increase the welfare of others, and its outcome behavior itself is intrinsically rewarding [37]. A typical component of altruism is the joy people experience in helping others [38]. Altruism can thus lead consumers to engage in contribution behaviors [39–41]. In the context of shared mobility, the object (e.g., bike) is accessed by all users, which suggests that a user's both in- and extra-role participations can be attributed to this kind of a user's intrinsic motivation. In other words, due to the fact that bike sharing is a service that is "what's mine is yours," users' required and/or voluntary behaviors often come from altruistic tendencies they have. Accordingly, with the understanding of the nature of altruism that aims to promote others' welfare, this study proposes the following hypotheses:

**Hypothesis 1a (H1a).** *Altruism has a positive effect on user in-role participation.*

**Hypothesis 1b (H1b).** *Altruism has a positive effect on user extra-role participation.*

To enhance customer value co-creation, firms usually offer incentives to their customers [31]. Obviously, most users expect rewards in return for their participation. Rewards can be divided into financial and non-monetary ones [42]. For bike sharing services, financial rewards are direct monetary payoffs, while non-monetary rewards are increases in user credit that would allow users to use the bikes at a lower rate. This study considers these two forms of rewards as a complete concept. Expected rewards, as extrinsic motivation, have

been demonstrated to enhance consumer positive behavior in virtual environments [43]. According to Yen et al. [22], rewards can lead to additional instrumental value for customers. Customers are thus more willing to cooperate with the service provider and engage in responsible and citizenship behavior [41]. Hence, this study anticipates that rewards can enhance user participation and formulate the following hypotheses accordingly:

**Hypothesis 2a (H2a).** *Rewards have a positive effect on user in-role participation.*

**Hypothesis 2b (H2b).** *Rewards have a positive effect on user extra-role participation.*

Opportunity-related factors can be both negative and positive [33]. Negative factors refer to situational elements that complicate and impede an opportunity, restricting a given behavior [34], while positive ones refer to elements that are conducive to achieving a desired outcome [36,44]. This study proposes that the perceived ease of use of the bike sharing system (technology) is critical in enhancing the opportunity to achieve the cooperation goals. According to Davis [45] and Meuter et al. [46], a system's ease of use refers to a system's having simple usage instructions and a straightforward process. A system that is easy to use can promote an individual's technology readiness, indicating the individual's propensity to embrace new technologies when completing tasks [47]. Balaji and Roy [32] confirmed that perceived ease of use has a positive effect on customer value co-creation as customer value co-creation behavior is primarily technology-enabled.

As bike users need to park and lock their bikes after use, if the system does not clearly inform them of recommended parking zones or the locks are not smart enough, their in-role participation could be negatively affected. In addition, if the system requires significant effort when reporting problems with the bikes, users would be less likely to engage in these behaviors. Therefore, as a situational factor, the system's perceived ease of use is assumed to minimize the effort required in participation. This reasoning leads to the following hypotheses:

**Hypothesis 3a (H3a).** *Perceived ease of use has a positive effect on user in-role participation.*

**Hypothesis 3b (H3b).** *Perceived ease of use has a positive effect on user extra-role participation.*

Ability refers to an individual's skills or knowledge in co-creating value with the service provider or other participants [35]. Customer knowledge is a key determinant of customer ability [34]. According to Lamberton and Rose [48], user knowledge refers to how familiar users are with and how much information they have about the commercial sharing program. While limited knowledge or experience may reduce a customer's processing ability [33], relevant knowledge has the potential to increase clarity regarding the user's role. According to Dong et al. [49], role clarity refers to a user's understanding of their roles and responsibilities when using the service. Users with a high level of knowledge will have a more precise understanding of what is required and even helpful for a successful transaction. Knowledge can also lead to self-efficacy, which enables a user to complete a specific behavior effectively [41]. That is, a bike sharing service user who has a high level of relevant knowledge not only knows well about "dos and don'ts," but also actively demonstrates voluntary behaviors. Accordingly, the following hypotheses are proposed:

**Hypothesis 4a (H4a).** *User knowledge has a positive effect on user in-role participation.*

**Hypothesis 4b (H4b).** *User knowledge has a positive effect on user extra-role participation.*

*2.4. Antecedents from SET*

SET has its roots in economics, emphasizing that interpersonal interactions and relationships are based on a cost-benefit exchange [50,51]. According to this theory, individuals

thus expect reciprocal benefits and reinforcement in their social interaction. Using this lens of SET, when consumer-brand (e.g., [52]) and consumer-firm relationships (e.g., [53]) are examined, brands and firms can be seen as social entities fostering relationships with customers. From a relationship quality perspective, firms strive to make their customers perceive relationship investment and quality in their customer-firm interactions, which could lead to positive changes in customer purchase behaviors [54]. According to Nyffenegger et al. [55], relationship quality can be divided into "cold" and "hot" dimensions. A "cold" relationship quality could be satisfaction that reflects the cumulative satisfaction a customer experiences when evaluating service performance relative to their expectations. A "hot" relationship quality could be customer (affective) commitment referring to the degree to which the customer is psychologically bonded to the service supplier based on favorable feelings it elicits [56,57]. This study investigates the impact of user satisfaction and user commitment formed through user–provider interactions on user participation in addition to the impact of rewards.

Customer participation is an example of social exchange behavior [14,58]. According to SET, customers with a high level of satisfaction and commitment are more likely to reciprocate their favorable treatment by engaging in contribution behaviors that are beneficial to the firm [54,59]. In the OCB research stream, researchers have applied SET to demonstrate that employee satisfaction and commitment can be strong predictors of employee OCBs and in-role task performance (e.g., [23]). In the context of bike sharing services, user satisfaction and user commitment may come from successful prior experiences of interacting with the service provider who endeavors to provide more value to users. Following SET, these enhanced user satisfaction and commitment have the potential to make users comply with relevant regulations and perform voluntary cooperative behaviors. Furthermore, in line with bike sharing users who play the role of "partial employee", this study suggests that user satisfaction and user commitment have the potential to increase user participation from the "user as a value co-creator" perspective [19,20]. Therefore, the following hypotheses are proposed:

**Hypothesis 5a (H5a).** *User satisfaction has a positive effect on user in-role participation.*

**Hypothesis 5b (H5b).** *User satisfaction has a positive effect on user extra-role participation.*

**Hypothesis 6a (H6a).** *User commitment has a positive effect on user in-role participation.*

**Hypothesis 6b (H6b).** *User commitment has a positive effect on user extra-role participation.*

According to SET [59] and the equity-first model [60], customers evaluate the service provider as being fair based on a comparison of the outcomes they wanted relative to inputs, which determines customer satisfaction. In this study, users who obtain rewards from service providers while performing required and voluntary cooperative behaviors will evaluate them as fair, which enhances user satisfaction. Moreover, following the cognitive–affective–conative framework [57], user satisfaction could lead to user commitment before influencing user behavior because the satisfying experiences could first result in favorable feelings. Taking these together with a focus on user–service provider interactions, this study proposes the following hypotheses:

**Hypothesis 7 (H7).** *Rewards have a positive effect on user satisfaction.*

**Hypothesis 8 (H8).** *User satisfaction has a positive effect on user commitment.*

On the basis of identifying user cooperation behavior as in-role participation and extra-role participation, this study adopts both the MOA model and SET in expounding the influences of factors from user–user, user–provider, and user–service interactions on

such two user participations. With discussions above, Figure 1 shows all the constructs and hypotheses as the research model of this study.

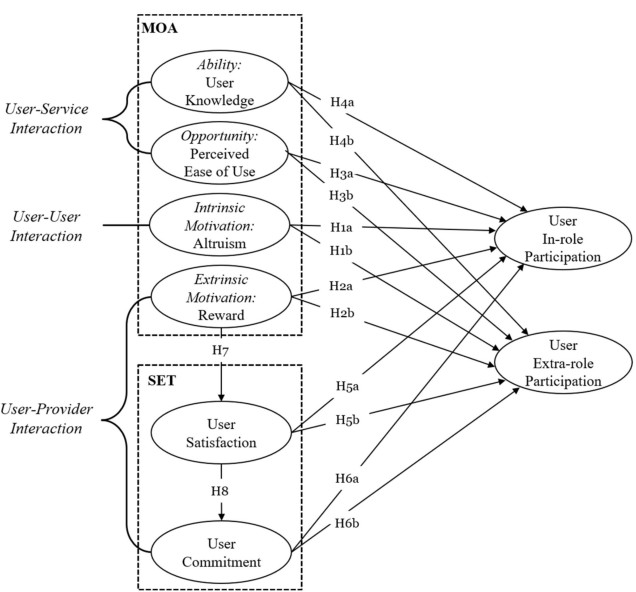

**Figure 1.** Research model.

## 3. Research Method

### 3.1. Measurements

For this study, a survey instrument was designed to obtain data on research variables. There are eight constructs, of which the measurement items were drawn from previous studies and slightly modified to ensure the appropriateness for this study. All of the constructs were measured with multiple items based on a seven-point Likert scale (1 = strongly disagree, 7 = strongly agree). The operational definitions of all the constructs, the measurement items, and the related sources are presented in Table 2.

**Table 2.** Operational definition and measurement items for each construct.

| Construct | Operational Definition and Items | Sources |
|---|---|---|
| Altruism | The degree to which a user enjoys doing something to benefit others | Cheung and Lee [39] |
| | 1. It feels good to help others. | |
| | 2. Helping others is pleasurable. | |
| | 3. Helping others is important to me. | |
| | 4. I like to help others. | |
| Reward | The extent to which a user can obtain monetary and credit score benefits from the service supplier | Yen et al. [22] |
| | 1. I can receive rewards from this service provider for my participation. | |
| | 2. This service provider offers me a variety of incentives for my participation. | |
| | 3. I can save money by participating. | |
| Perceived Ease of Use | The extent to which a user perceives that using the commercial bike sharing system requires little effort | Venkatesh [61] |
| | 1. This system is clear and understandable. | |
| | 2. Interacting with this system does not require a lot of mental effort. | |
| | 3. I find this system to be easy to use. | |
| | 4. I find it easy to get this system to do what I want it to do. | |

**Table 2.** *Cont*.

| Construct | Operational Definition and Items | Sources |
|---|---|---|
| User Knowledge | The degree to which a user is knowledgeable about the commercial bike sharing service | Lamberton and Rose [48] |
| | 1. I am familiar with this commercial bike sharing service. | |
| | 2. I have experience with this commercial bike sharing service. | |
| | 3. I know much about how this commercial bike sharing system works. | |
| User Satisfaction | The degree to which a user is satisfied with his or her previous interaction experience with the commercial bike sharing service provider | Wangenheim and Bayón [59] |
| | 1. My experiences with this service have always been pleasant. | |
| | 2. Based on all my experiences with this service, I am very satisfied. | |
| | 3. Overall, I am very satisfied with the service of this commercial bike sharing system provider. | |
| User Commitment | The extent to which a user is psychologically bonded with the commercial bike sharing service provider | Hennig-Thurau et al. [40]; Gruen et al. [56] |
| | 1. I feel a strong sense of belonging to this provider. | |
| | 2. I have a strong emotional attachment with this provider. | |
| | 3. This provider's service has a great deal of personal meaning for me. | |
| | 4. My relationship with this provider is very important to me. | |
| In-Role Participation | The degree to which a user performs activities that are expected and required | Yen et al. [22]; Yi and Gong [24] |
| | 1. I adequately complete all the expected actions. | |
| | 2. I perform all the required tasks. | |
| | 3. I meet the formal performance requirements. | |
| | 4. I fulfill my responsibilities to this business. | |
| Extra-Role Participation | The extent to which a user performs voluntary activities that are helpful to the service provider | Yen et al. [22]; Yi and Gong [24] |
| | 1. I provide helpful feedback to this service provider. | |
| | 2. If I have a useful idea on how to improve this service, I let this service provider know. | |
| | 3. When I experience a problem, I let this service provider know about it. | |
| | 4. I often offer this service provider useful information (e.g., bike defects). | |

### 3.2. Data Collection

This study used an online survey method for data collection. The unit of analysis was a user who has at least one experience of commercial bike sharing services in China. The online survey was conducted via Sojump (www.sojump.com), which is a popular Chinese survey website. A virtual snowball sampling technique was employed to share and forward the survey links on WeChat, a dominant Chinese social networking service, which can help collect data from a general population in China. According to Baltar and Brunet [62], a virtual snowball sampling technique is suitable for data collection in the social media era, although it may cause some concerns for community biases, non-random sampling errors, or lacks of control. Further, in order to attract more attention from respondents, 1–2 CNY rewards were randomly provided to them. The period of data collection lasted about one month (from January 2018 to February 2018). Respondents were first requested to report the most frequently used bike sharing service as well as the duration of use on the opening page of the questionnaire. A total of 495 questionnaires were submitted, of which 438 valid and complete ones were acquired. Out of the 495 questionnaires collected, 57 were excluded due to incomplete responses with missing the service name or the duration of use, or aberrant responses lacking justification. The respondents of the discarded questionnaires were not statistically different from those of the included questionnaires regarding the

sample demographics. In addition, the service names of bike sharing such as Mobike and ofo, which explain about 90 percent (89.7%) of the sample in the study, were the top two bike sharing services in China during the period of data collection. Demographic information of the sample is shown in Table 3.

**Table 3.** Demographics of Respondents (*n = 438*).

| Category | | Frequency | Percent (%) |
|---|---|---|---|
| Gender | Male | 243 | 55.5 |
| | Female | 195 | 44.5 |
| Age | <20 | 48 | 11.0 |
| | 20–29 | 246 | 56.2 |
| | 30–39 | 122 | 27.9 |
| | >39 | 22 | 5.0 |
| Education | High school or lower | 74 | 16.9 |
| | Bachelor's or college degree | 274 | 62.6 |
| | Graduate degree | 90 | 20.5 |
| Income (Monthly, CNY) | <5000 | 175 | 40.0 |
| | 5001–10,000 | 153 | 34.9 |
| | 10,001–15,000 | 73 | 16.7 |
| | 15,001–20,000 | 17 | 3.9 |
| | >20,000 | 20 | 4.6 |
| Duration of Use | <6 months | 132 | 30.1 |
| | 6–12 months | 226 | 51.6 |
| | 13–18 months | 53 | 12.1 |
| | >18 months | 27 | 6.2 |
| Most Frequently Used Bike Sharing Service | Mobike | 202 | 46.1 |
| | ofo | 191 | 43.6 |
| | Others | 45 | 10.3 |
| Total | - | 438 | 100 |

## 4. Data Analysis and Results

### 4.1. Measurement Model

This study utilized the partial least squares structural equation model (PLS-SEM) method to conduct a confirmatory factor analysis (CFA), evaluating both reliability and validity of the measurement. As shown in Table 4, the values of composite reliability (CR) and Cronbach's α for all constructs were higher than the threshold value of 0.7, suggesting acceptable scale reliability and internal consistency [63,64]. Regarding convergent validity, in addition to the standardized factor loadings of indicators for all constructs being significantly greater than 0.7, the values of average variance extracted (AVE) for all constructs exceeded the recommended minimum of 0.5, confirming acceptable convergent validity [63,64]. Moreover, following Fornell and Larcker [63], this study compared the square root of AVE for each construct with the inter-construct correlation estimates to check discriminant validity. Table 5 reports the square roots of AVE (the diagonal elements in bold) for constructs and construct correlation estimates. Each square root of AVE was greater than its corresponding row and column elements, indicating acceptable discriminant validity of the instruments.

Further, self-reported data from a single source may have a common method bias (CMB), which threatens the validity of the study. Following Liang et al. [65], the unmeasured latent method construct (ULMC) approach in PLS was used to assess the level of the CMB. The results revealed that the average substantively explained variance of indicators was 0.754, whereas the average method-based variance of the indicators was 0.005. The ratio of substantive variance to method variance was considerably large (150.8:1), and most of the method factor loadings were also insignificant. Thus, the small magnitude and insignificance of method variance indicate that the CMB is not a significant problem in this study.

**Table 4.** Results of reliability and convergent validity tests.

| Construct | Indicators | Standardized Factor Loadings | Cronbach's $\alpha$ | CR | AVE |
|---|---|---|---|---|---|
| Altruism | ALT1 | 0.909 | 0.892 | 0.925 | 0.755 |
| | ALT2 | 0.871 | | | |
| | ALT3 | 0.811 | | | |
| | ALT4 | 0.883 | | | |
| Reward | REW1 | 0.900 | 0.865 | 0.917 | 0.787 |
| | REW2 | 0.907 | | | |
| | REW3 | 0.854 | | | |
| Perceived Ease of Use | PEU1 | 0.860 | 0.871 | 0.912 | 0.757 |
| | PEU2 | 0.835 | | | |
| | PEU3 | 0.858 | | | |
| | PEU4 | 0.844 | | | |
| User Knowledge | UK1 | 0.877 | 0.829 | 0.898 | 0.745 |
| | UK2 | 0.844 | | | |
| | UK3 | 0.868 | | | |
| User Satisfaction | US1 | 0.895 | 0.854 | 0.911 | 0.774 |
| | US2 | 0.867 | | | |
| | US3 | 0.877 | | | |
| User Commitment | UC1 | 0.868 | 0.893 | 0.926 | 0.757 |
| | UC2 | 0.865 | | | |
| | UC3 | 0.855 | | | |
| | UC4 | 0.892 | | | |
| In-Role Participation | IP1 | 0.867 | 0.906 | 0.934 | 0.780 |
| | IP2 | 0.894 | | | |
| | IP3 | 0.868 | | | |
| | IP4 | 0.901 | | | |
| Extra-Role Participation | EP1 | 0.850 | 0.855 | 0.902 | 0.697 |
| | EP2 | 0.817 | | | |
| | EP3 | 0.813 | | | |
| | EP4 | 0.858 | | | |

**Table 5.** Results of Correlation Analysis and Discriminant Validity Tests.

| Construct | Mean | S.D. | 1 | 2 | 3 | 4 | 5 | 6 | 7 | 8 |
|---|---|---|---|---|---|---|---|---|---|---|
| 1. ALT | 5.749 | 1.049 | **0.869** | | | | | | | |
| 2. REW | 5.358 | 1.247 | 0.543 | **0.887** | | | | | | |
| 3. PEU | 5.623 | 0.944 | 0.690 | 0.467 | **0.849** | | | | | |
| 4. UK | 5.397 | 0.992 | 0.631 | 0.499 | 0.694 | **0.863** | | | | |
| 5. US | 5.549 | 0.955 | 0.692 | 0.547 | 0.711 | 0.635 | **0.880** | | | |
| 6. UC | 5.127 | 1.015 | 0.541 | 0.516 | 0.504 | 0.633 | 0.636 | **0.870** | | |
| 7. IP | 5.862 | 1.083 | 0.716 | 0.514 | 0.739 | 0.644 | 0.644 | 0.485 | **0.883** | |
| 8. EP | 5.255 | 0.949 | 0.620 | 0.556 | 0.552 | 0.597 | 0.649 | 0.563 | 0.542 | **0.835** |

ALT: Altruism, REW: Reward, PEU: Perceived Ease of Use, UK: User Knowledge, US: User Satisfaction, UC: User Commitment, IP: In-Role Participation, EP: Extra-Role Participation. The diagonal numbers in **bold** are the square roots of the AVE.

### 4.2. Structural Model

SmartPLS (v 3.0, SmartPLS GmbH, Germany, 2014.) was utilized to perform a path analysis to test the research hypotheses. Figure 2 depicts the results of hypotheses test. First, among motivation, opportunity, and ability antecedents, altruism was found to have positive impacts on in- and extra-role participations ($\beta = 0.306$, $p < 0.001$; $\beta = 0.196$, $p < 0.001$, respectively), supporting H1a and H1b. Rewards were found to have positive effects on in- and extra-role participations ($\beta = 0.097$, $p < 0.05$; $\beta = 0.181$, $p < 0.001$, respectively), supporting H2a and H2b. Perceived ease of use was found to have a positive influence on in-role participation ($\beta = 0.368$, $p < 0.001$), but had no effect on extra-role participation ($\beta = -0.018$, $p > 0.05$), which suggests that H3a was supported and H3b was not. Meanwhile, user knowledge was found to have positive effects on in- and extra-role participations ($\beta = 0.136$, $p < 0.01$; $\beta = 0.169$, $p < 0.01$, respectively), supporting H4a and H4b.

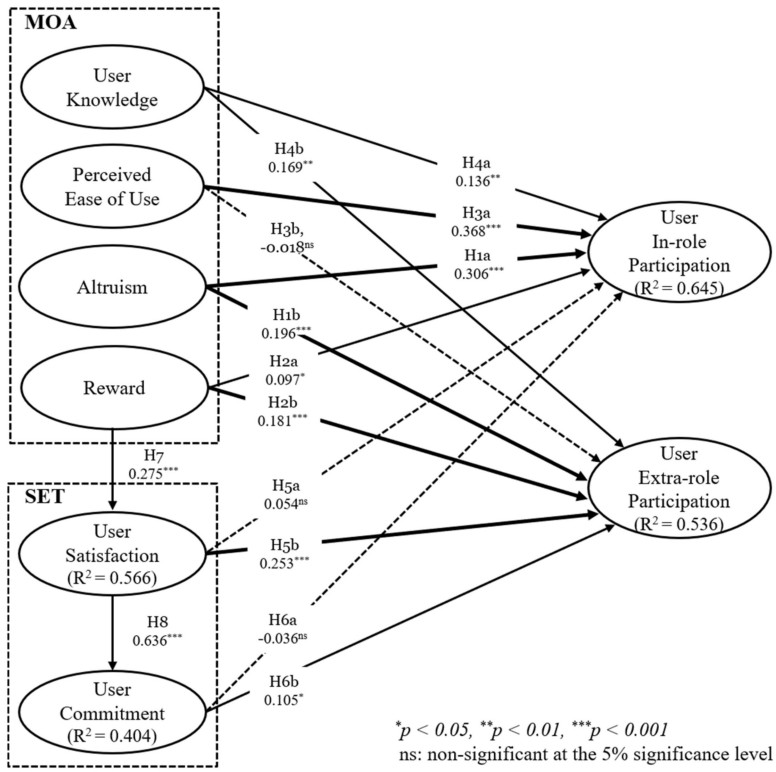

**Figure 2.** Results of Hypotheses test.

Second, user satisfaction and user commitment were both found to have positive impacts on user extra-role participation ($\beta = 0.253$, $p < 0.001$; $\beta = 0.105$, $p < 0.05$, respectively), supporting H5b and H6b. However, they were found to have no effect on user in-role participation ($\beta = 0.054$, $p > 0.05$; $\beta = -0.036$, $p > 0.05$, respectively), which suggests that H5a and H6a were not supported. Finally, rewards were found to have a positive effect on user satisfaction ($\beta = 0.275$, $p < 0.001$), and user satisfaction was found to have a positive influence on user commitment ($\beta = 0.636$, $p < 0.001$), supporting both H7 and H8.

## 5. Discussion

### 5.1. Discussion of Findings

The results provide evidence that most user–user, user–provider, and user–service interaction antecedents can effectively predict user in- and extra-role participations with further theoretical background of the MOA model and SET. Interestingly, it was found that the perceived ease of use cannot enhance user extra-role participation. Commercial bike sharing services are facilitated by smart self-service technologies, aiming to provide maximum convenience. Therefore, a system that is easy to use directly contributes to the saving of time and effort [45,61], which in turn has the potential to eliminate violations caused by time pressure. On the other hand, a system that is easy to use is helpful for users in overcoming the uncertainty concerning the required behavior [46]. Therefore, perceived ease of use was found to have a significant effect on in-role participation. User extra-role participation reflects user voluntary behavior, which implies that time pressure as well as concerns regarding time and effort may be absent when a user decides to engage in these behaviors. In other words, if a user decides to engage in these voluntary behaviors, s/he knows it would take time and effort, and therefore, the impact of the system's perceived ease of use might weaken. Hence, perceived ease of use was found not to be a necessary opportunity antecedent of user extra-role participation.

This study also found that user satisfaction and user commitment do not increase user in-role participation. Organ and Ryan [23] argued that the relationship between job satisfaction and OCB is stronger than that between satisfaction and in-role task performance. Furthermore, regarding customer in-role participation, as Groth [21] stated, "to successfully complete a transaction at a bank, a customer must still fill out the necessary forms, regardless of his or her satisfaction during the service encounter" (p. 12), both user satisfaction and user commitment were found to have significant impacts only on user extra-role participation.

### 5.2. Theoretical Implications

This study firstly contributes to the existing literature on value co-creation and CCB by examining user cooperation behaviors as cases of both in-role and extra-role participations in the context of shared mobility services. This study proposes that user in-role participation (responsible cooperation behavior) and extra-role participation (voluntary cooperation behavior) should both be emphasized in research on the sharing economy. This emphasis would also expand research on consumer (user) behavior especially in the sharing economy era.

Second, to better understand approaches to promoting user participation, this study considers the particularities of shared mobility services in the sharing economy [2,3,10,11], and comprehensively identifies the key drivers of user participation from the perspectives of user–user, user–provider, and user–service interactions. It is worth noting that research on why individuals adopt the sharing economy is helpful for predicting and enhancing users' sharing economy propensity [48,66], while research on consumer/user (i.e., individuals who have adopted and are using the sharing economy platforms) behaviors, which includes negative and cooperative behaviors [9–12,67], contributes to overcoming the deficiencies of the sharing economy. This study, which proposes a theoretical framework for enhancing user participation in bike sharing services, belongs to the latter field of research.

Third, this study further empirically examines the relationships between user participation and MOA- and relationship quality-related antecedents by applying and integrating the MOA model and SET. Following [33], MOA-related antecedents that include individual and situational factors are more comprehensive in predicting user participation in the context of shared mobility. At the same time, relationship quality-related antecedents (i.e., user satisfaction and user commitment) enhance user extra-role participation, which further emphasizes the importance of CRM [52,55,68] in the shared mobility as well as the sharing economy research streams. Overall, the drivers of user participation are enriched, which goes beyond the focus on motivation and expected benefits (e.g., [69,70]).

Lastly, Bardhi and Eckhardt [9] suggested that the "big brother" model of governance could be used to prevent users from engaging in opportunistic behavior. This solution is based on the fact that shared products are accessed rather than owned by customers, thus requiring governance and sanctions. However, Hartl et al. [11] argued that governance systems sometimes fail to increase customer cooperation because they are likely to diminish user self-determination and initiative. This study not only stresses the importance of user self-determination and initiative but also contributes to understanding solutions for alleviating user opportunistic behaviors as well as the negative impact they can bring about.

### 5.3. Practical Implications

In addition to theoretical implications, this study makes several contributions to practice. First, for sustainable development, bike sharing service firms should endeavor to stimulate user in- and extra-role participations simultaneously. Most access-based service providers use deposit and penalty methods to ensure user in-role participation. However, these approaches should be used with great caution to encourage extra-role participation, and in some cases, have no effect on in-role participation [11]. Research findings imply that providing rewards is an effective way to strengthen both user in- and extra-role participations. For example, a commercial bike sharing firm, Hellobike, obtains support from Alibaba's Ant Financial Service to implement a credit-scoring system, providing users with direct monetary incentives to encourage them to use the bikes appropriately and report issues to the firm. It is worth noting that offering money requires a large amount of funds, and should, therefore, be used only in combination with other tactics. The research findings also indicate that user altruism could strongly influence user in- and extra-role participations despite the "what's mine is yours" mindset of shared mobility consumption. Therefore, firms should empower their customers and fully stimulate their initiative and altruistic considerations through reciprocal appeals.

Second, the service of commercial bike sharing is a smart technology-based self-service. Hence, firms should make their system smart enough to save users' time and effort when using the service. For example, firms should lay out as many appropriate parking zones as possible. As user knowledge can increase through training, firms should also frequently share relevant useful information with users (especially new users) via their mobile applications or by inviting users to participate in events and campaigns.

Third, this study provides evidence that bike sharing firms should pay attention to building strong relationships with their customers. The loyal user is shown to be more likely to be involved in extra-role participation, implying that enhancing the user–provider relationship is a cost-efficient way to lead users to engage in citizenship behavior. Thus, firms should improve their data analysis and bike redistribution capabilities to promote bike accessibility, ensuring that bikes are always available [71]. As vehicle damage caused by normal wear and tear or user misbehavior is another constraint for bike availability, firms should promptly respond to user feedback and maximize bike availability. In short, commercial bike sharing firms should continuously strive to improve service quality [72]. Bike sharing firms that effectively design customer loyalty program tactics to provide users with different kinds of rewards could also promote user satisfaction.

### 5.4. Limitations and Future Research Directions

This study, viewing users as active actors in shared mobility consumption, addressed the key factors that could enhance user participation. In practice, commercial sharing service suppliers still use the big-brother model of governance to prevent user negative behaviors. However, due to psychological reactance [73], users may feel this model to be a threat to their freedom and may hesitate to participate. Future research that applies neutralization and deterrence theories [74] to investigate the effects of firm marketing activities on user in- and extra-role participations would provide further insight on this topic.

Karpen et al. [75] proposed the S-D orientation to address the fact that firms should have individuated, relational, ethical, empowered, developmental, and concerted interaction capabilities to best facilitate and enhance value co-creation. For a holistic approach to enhance user cooperation in the context of the sharing economy, it is recommended that future research explores the effect of these six capabilities on user psychological and behavioral responses.

Finally, it should be also noted that there exist some sampling related issues in the study. This study used a virtual snowball sampling method to collect data. As this technique may cause some issues of community bias, non-random sampling error, or lack of control, future research that uses field survey data collected by random sampling is recommended to increase the generalizability of the study. Since the sample of the study was relatively small and somewhat outdated, future research should include larger samples with more recent cases to sufficiently validate the findings of this research.

**Author Contributions:** Conceptualization, L.L. (Liguo Lou), J.K., and S.-B.Y.; methodology, L.L. (Liguo Lou) and L.L. (Lin Li); software, L.L. (Liguo Lou); validation, L.L. (Liguo Lou), L.L. (Lin Li), and S.-B.Y.; formal analysis, L.L. (Liguo Lou); investigation, L.L. (Liguo Lou) and L.L. (Lin Li); resources, L.L. (Liguo Lou); data curation, J.K. and S.-B.Y.; writing—original draft preparation, L.L. (Liguo Lou); writing—review and editing, J.K., L.L. (Lin Li), and S.-B.Y.; visualization, L.L. (Liguo Lou); supervision, J.K.; project administration, J.K. and S.-B.Y.; funding acquisition, S.-B.Y. All authors have read and agreed to the published version of the manuscript.

**Funding:** This work was supported by the Ministry of Education of the Republic of Korea and the National Research Foundation of Korea (NRF-2020S1A5B8103855).

**Institutional Review Board Statement:** Not applicable.

**Informed Consent Statement:** Not applicable.

**Data Availability Statement:** The data presented in this study are available on request from the corresponding author.

**Conflicts of Interest:** The authors declare no conflict of interest.

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
