# Peer review of "Promoting User Participation of Shared Mobility in the Sharing Economy: Evidence from Chinese Bike Sharing Services"

_sustainability, doi:10.3390/su13031533_

Round 1
Reviewer 1 Report
The paper investigated on the influences of perceptive constructs (such as altruism, reward, ease of use) to the user participation behavior (responsible cooperation behavior and voluntary cooperation behavior). User participation is a timely and important research in the field of Bike-Sharing Service especially on incentivizing the participants. The authors are certainly tackling an interesting research area and provided substantial effort on framing the proposed framework. I believe the paper is acceptable as is condition.
Author Response
We thank you for your positive evaluation. Meanwhile, we have tried our best to address all of the concerns that other two reviewers raised.
Comment: The paper investigated on the influences of perceptive constructs (such as altruism, reward, ease of use) to the user participation behavior (responsible cooperation behavior and voluntary cooperation behavior). User participation is a timely and important research in the field of Bike-Sharing Service especially on incentivizing the participants. The authors are certainly tackling an interesting research area and provided substantial effort on framing the proposed framework. I believe the paper is acceptable as is condition.
Response: We thank you for your positive feedback. We have tried our best to address all of the concerns that the reviewer term raised. The followings are the point-by-point responses to the comments, and now we report:
- Shared mobility has been clearly defined and better demonstrated in the “1. Introduction” section. Also, this term has been used across the revised manuscript including the title to narrow down the focus of this study onto shared transportation services.
- The representativeness issue of the sample collected from this study has been addressed not only by strengthening the rationale of adopting a snowball sampling technique with some concerns, but also by demonstrating the non-difference of the demographic characteristics between included and excluded respondents in the “3.2 Data Collection” section.
- Additional resources and references suggested by the review team have been added in the revised manuscript.
- The clear statement of the research purpose and a new paragraph of the structure of this paper have been added at the end of the “1. Introduction” section.
- The definition, role, and advantage/disadvantage of the bike-sharing service have been discussed in more details in the “1. Introduction” section.
- The table that was located in Appendix has been relocated in the “3.1 Measurement” section as Table 2. Also, more descriptive information about the survey questionnaire and procedure have been presented in both the “3.1 Measurement” and “3.2 Data Collection” sections.
- We have tried our best to avoid using personal languages throughout the manuscript.
Reviewer 2 Report
This paper uses a structural equation model to examine the behavior of bikesharing users in China. A few questions/comments for the authors:
The introduction does not clearly call out shared mobility (which the authors refer to as access based services). The authors may consider using shared mobility and/or on-demand mobility, which is a more recognized international term. This would also help to more narrowly focus the discussion around shared transportation services (rather than the sharing economy more broadly) that can have different behaviors and motivating factors.
Methods: Please clarify if there was a geographic scope of the survey data collection (e.g., specific operators or cities) or a general population in China. If a general population, is a sample less than 500 respondents sufficient to draw conclusions on motivations? How do the demographics of respondents compare to the general population from where the sample was collected? Were responses weighted based on the general population distribution? What are the limitations of the online snowball technique? Although the limitations are discussed at the very end, these should be discussed in the methodological section with some discussion about how these were addressed/controlled for, if at all possible.
Some additional resources that may be helpful to consider is revising this paper:
https://rosap.ntl.bts.gov/view/dot/50553
https://escholarship.org/uc/item/9678b4xs
Author Response
We would like to thank you for the excellent and constructive feedback. As described in this response, we have made significant changes to the paper to address all of the concerns raised by you. We believe that our manuscript has been much improved as a result of this revision, hoping that you concur with our revised paper. We have also highlighted the changes within the revised manuscript by using red-colored texts. The point-by-point responses to your comments (starts with ‘AR#,’ which means ‘answer to R#’) will follow immediately after this note.
R2. 1. This paper uses a structural equation model to examine the behavior of bikesharing users in China. A few questions/comments for the authors:
AR2. 1. We thank you for your affirmative feedback, which means a lot to us as researchers. We have tried our best to address all of the concerns you raised. The followings are the point-by-point responses to the comments, and we hope our revision meet your requirements and standards.
R2. 2. The introduction does not clearly call out shared mobility (which the authors refer to as access based services). The authors may consider using shared mobility and/or on-demand mobility, which is a more recognized international term. This would also help to more narrowly focus the discussion around shared transportation services (rather than the sharing economy more broadly) that can have different behaviors and motivating factors.
AR2. 2. Thank you for the constructive suggestion. We believe this comment is critical for the quality of the paper. So, we have tried our best to make this concept (i.e., shared mobility) and related discussions appear in our revised manuscript in a clearer manner. Currently, shared mobility has been clearly defined and better demonstrated in the Introduction section. Also, this term has been consistently used across the revised manuscript including the title to narrow down the focus of this study onto shared transportation services.
On pages 1 and 2, in the “1. Introduction” section:
Shared mobility is a representative example of the sharing economy, which refers to the shared use of bikes, motorcycles, cars, or other transportation vehicles, contributing to a number of social, environmental, and transportation cost-related benefits (Shaheen et al. 2019). Among these shared mobility services, the commercial bike sharing service has grown to become a sustainable transportation alternative for millions across the globe, providing a low-carbon solution to the ‘last mile’ problem (Fishman et al. 2014; Nogal and Jiménez 2020; Wang et al. 2019). As Shaheen et al. (2019) suggested, the bike sharing service primarily comprises of three models: station-based, dock-less, and hybrid models. In a station-based model, users are required to return bikes to the bike stations, while in a dock-less model, users can pick up and return them within a predefined geographic region. A hybrid model integrates the characteristics of both the two models. In China, most bike sharing services have been adopting a mobile technology-based dock-less model that enables users to pick up and return bikes anywhere within an operating zone via mobile applications. A coin has two sides and so does the dock-less model; this free-floating bike sharing system enhances user flexibility and convenience, but at the same time, it gives rise to high operating costs with respect to re-allocating bikes for users on demand, monitoring users’ misbehaviors, fixing bikes, and so forth. Hello Bike is a firm that survives in a fierce competition in China (van Waes et al. 2020). In order to improve its business performance, Hello Bike has implemented a number of user-centered marketing activities, such as providing membership cards to users, employing customer relationship management programs, and encouraging users to report bike defects and/or other users’ violations. Moreover, big data analytics has made it possible to clearly inform users of proper parking zones for pick-ups and returns and minimize the operating cost by allowing users to do the tasks employees have to do.
Bike sharing, as one representative type of shared mobility, is essentially an access-based service that has transformed the understanding of possession/ownership and has reminded the importance of individual obligations that underlie shared objectives. Access-based services refer to services that allow users to access a resource for a defined period of time in return for an access payment while the service provider retains full legal ownership of the resource (Bardhi and Eckhardt 2012; Schaefers et al. 2016).
R2. 3. Methods: Please clarify if there was a geographic scope of the survey data collection (e.g., specific operators or cities) or a general population in China. If a general population, is a sample less than 500 respondents sufficient to draw conclusions on motivations? How do the demographics of respondents compare to the general population from where the sample was collected? Were responses weighted based on the general population distribution? What are the limitations of the online snowball technique? Although the limitations are discussed at the very end, these should be discussed in the methodological section with some discussion about how these were addressed/controlled for, if at all possible.
AR2. 3. Thank you for your valuable comments. As per your suggestions, the representativeness issue of the sample collected from this study has been addressed not only by strengthening the rationale of adopting a snowball sampling technique with some concerns, but also by demonstrating the non-difference of the demographic characteristics between included and excluded respondents in the “3.2 Data Collection” section. Also, more detailed discussions regarding the concern of using the snowball sampling technique have been added in the “5.4 Limitations and Future Research Directions” section. As we have tried our best to address these concerns, we hope that you can concur with our revision.
On pages 9 and 10, in the “3.2 Data Collection” section:
This study used an online survey method for data collection. The units of analysis were commercial bike sharing service users in China. The online survey was conducted via Sojump (www.sojump.com), which is a popular Chinese survey website. A virtual snowball sampling technique was employed to share and forward the survey links on WeChat, a dominant Chinese social networking service, which can help collect data from a general population in China. According to Baltar and Brunet (2012), a virtual snowball sampling technique is suitable for data collection in the social media era, although it may cause some concerns for community biases, non-random sampling errors, or lacks of control. Further, in order to attract more attention from respondents, 1-2 CNY rewards were randomly provided to them. The period of data collection lasted about one month (from January 2018 to February 2018). Respondents were first requested to report the most frequently used bike sharing service as well as the duration of use on the opening page of the questionnaire. A total of 495 questionnaires were submitted of which 438 valid and complete ones were acquired. Out of the 495 questionnaires collected, 57 were excluded due to incomplete responses with missing service brand name or duration of use, or aberrant responses lacking justification. The respondents of the discarded questionnaires were not statistically different from those of the included questionnaires regarding the sample demographics. In addition, the service names of bike sharing such as Mobike and ofo, which explain about 90 percent (89.7%) of the sample in the study, were the top two biggest and representative bike sharing services in China during the period of data collection. Demographic information of the sample is shown in Table 3.
On pages 14 and 15, in the “5.4 Limitations and Future Research Directions” section:
Finally, it should be also noted that there exist some sampling related issues in the study. This study used a virtual snowball sampling method to collect data. As this technique may cause some issues of community bias, non-random sampling error, or lack of control, future research that uses field survey data collected by random sampling is recommended to increase the generalizability of the study. Since the sample of the study was relatively small and somewhat outdated, future research should include larger samples with more recent cases to sufficiently validate the findings of this research.
R2. 4. Some additional resources that may be helpful to consider is revising this paper:
https://rosap.ntl.bts.gov/view/dot/50553
https://escholarship.org/uc/item/9678b4xs
AR2. 4. We thank you for your helpful suggestions. Additional resources have been very helpful to revise this paper and enrich our knowledge about this topic for the follow-up research. We have cited the secondly suggested paper in the revised manuscript as follows. Again, we thank you for the opportunity to revise the paper.
At the end of page 1:
Shared mobility is a representative example of the sharing economy, which refers to the shared use of a bicycle, motorcycle, car, or other transportation mode, contributing to a number of social, environmental, and transportation cost-related benefits (Shaheen et al. 2019).
At the beginning of page 2:
As Shaheen et al. (2019) suggested, the biking sharing service primarily comprises of three models: station-based, dock-less, and hybrid models. In a station-based model, users are required to return bikes to the bike stations, while in a dock-less model, users can pick up and return them within a predefined geographic region. A hybrid model integrates the characteristics of both the two models.

Reviewer 3 Report
Interesting paper. In the reviewed paper, Authors presented the problem of promoting user participation of access-based services in the sharing economy. It was evidence from Chinese bike-sharing platforms. This paper classifies user participation behavior into two different types, i.e. in-and extra-role participations, integrates the motivation-opportunity-ability model and social exchange theory in order to identify key antecedents, and empirically examines the influences of user-user, user-provider, and user-service interaction-related factors on user participation in the context of access-based services. In my opinion, paper can be published, after taking into account the following remarks:
- at the end of the Introduction section, Authors should clearly state what was the aim of the article and what parts the article consists of, briefly describing what was contained in each section,
- in the article, the Authors deal with the bike-sharing system, as the paper research area. Despite this, the Authors did not define this system, from the point of view of an engineering system, i.e., describing it's role, its advantages and disadvantages. Authors should add one short paragraph in the Introduction section devoted to describing this type of system, at the same time Authors should mention some important papers publishing last time in the Sustainability journal regarding bike-sharing problems, such are as follows: https://doi.org/10.3390/su12219062, https://doi.org/10.3390/su12083285,
- https://doi.org/10.3390/su12156124,
- on the "Figure 1. Research Model" the legend should be added, because on the Figure are acronyms "H1a, H1b, H2a, H2b, H3a, H3b, etc." without explaining their meaning. In fact, these hypotheses are described in the paper text, but for readers, their meanings should also be explained on the figure,
- It would be better to not divide section 3 into further subsections, i.e. "3.1 Measurements" and "3.2. Data Collection", because their content is too short. In this case, it would be better to contain all in section 3 (without further divisions),
- in the text of the article, the formal language should be used, using impersonal phrases, i.e. "done" (instead of we did), "prepared" (instead of we prepared), "worked out" (instead of we developed), etc. So, please correct these phrases throughout the article, e.g. line 302 ... "For our study" ..., line 319, ..."We utilized "..., etc.,
- information about survey research presented in section 3 should be presented in a more detailed way,
- line 313-315-Authors wrote that ..."313495 questionnaires were submitted of which 438 valid and complete samples were acquired. Many of questionnaires were not included further in the study because they were not valid and complete. Could Authors develop this description of rejected questionnaires? In addition, in the Discussion section, Authors wrote about the advantages of the "the snowball sampling method", but based on the collected data, it can be seen that a lot of surveys have been rejected. Authors should provide comments in this area,
- Appendix "Operational Definition and Measurement Items for Each Construct"'should be included in section 3. Moreover, more descriptive information about this questionnaire should be provided.
Author Response
Responses to the Comments of Reviewer #3
We would like to thank you for the excellent and constructive feedback. As described in this response, we have made significant changes to the paper to address all of the concerns raised by you. We believe that our manuscript has been much improved as a result of this revision, hoping that you concur with our revised paper. We have also highlighted the changes within the revised manuscript by using red-colored texts. The point-by-point responses to your comments (starts with ‘AR#,’ which means ‘answer to R#’) will follow immediately after this note.
R3. 1. Interesting paper. In the reviewed paper, Authors presented the problem of promoting user participation of access-based services in the sharing economy. It was evidence from Chinese bike-sharing platforms. This paper classifies user participation behavior into two different types, i.e. in-and extra-role participations, integrates the motivation-opportunity-ability model and social exchange theory in order to identify key antecedents, and empirically examines the influences of user-user, user-provider, and user-service interaction-related factors on user participation in the context of access-based services. In my opinion, paper can be published, after taking into account the following remarks:
AR3. 1. We thank you for your affirmative feedback. We have tried our best to address all of the concerns that you raised. The followings are the point-by-point responses to the comments, and we hope our revision meet your requirements and standards.
R3. 2. At the end of the Introduction section, Authors should clearly state what was the aim of the article and what parts the article consists of, briefly describing what was contained in each section.
AR3. 2. We thank you for your helpful feedback. We totally agree with your comments and have revised our manuscript accordingly. Looking carefully into this review point, the clear statement of the research purpose and a new paragraph of the structure of the paper have been added into the “1. Introduction” section.
On pages 2 to 3, in the “1. Introduction” section:
In an attempt to contribute to the exploration of user participation in access-based services, this study classifies user participation into two different types (i.e., user in-role and extra-role participations) based on the discussion of value co-creation and customer citizenship behavior (CCB). In addition, drawing upon the motivation-opportunity-ability (MOA) model and social exchange theory (SET), this study aims to identify several key antecedent factors (i.e., altruism, rewards, perceived ease of use, user knowledge, user satisfaction, and user commitment) and validate the specific and different mechanisms to enhance user participation in the context of access-based services. Continuing on the path established by Schaefers et al. (2016), Park and Armstrong (2017), and Eckhardt et al. (2019) in the specific context of shared mobility, this study focuses on user-user, user-provider, and user-service interactions to identify key factors that have the potential to enhance two different types of user participation. To empirically examine these relationships, this paper focuses on the commercial bike sharing services in China as this industry has seen a rise, fall, and resurgence in the span of only a few years, during which the two types of user participation play critical roles in determining business performance.
The rest of the paper is organized as follows. Section 2 presents conceptual background and hypotheses development corresponding to the research questions, respectively. The research method is discussed in Section 3, while the results of data analyses and hypotheses tests are presented in Section 4. Lastly, the study findings, implications for both theory and practice, and limitations with future research directions are discussed in Section 5.
R3. 3. In the article, the Authors deal with the bike-sharing system, as the paper research area. Despite this, the Authors did not define this system, from the point of view of an engineering system, i.e., describing it's role, its advantages and disadvantages. Authors should add one short paragraph in the Introduction section devoted to describing this type of system, at the same time Authors should mention some important papers publishing last time in the Sustainability journal regarding bike-sharing problems, such are as follows: https://doi.org/10.3390/su12219062,
https://doi.org/10.3390/su12083285,
https://doi.org/10.3390/su12156124,
AR3. 3. Thank you for your helpful comments. We concur with your point. Accordingly, the definition, role, and advantage/disadvantage of the bike-sharing service have been discussed in more details in the “1. Introduction” section. In addition, all of the three related papers you suggested have been cited in the revised manuscript as well. We hope that we have appropriately addressed this review point.
On pages 1 and 2, in the “Introduction” section:
Shared mobility is a representative example of the sharing economy, which refers to the shared use of bikes, motorcycles, cars, or other transportation vehicles, contributing to a number of social, environmental, and transportation cost-related benefits (Shaheen et al. 2019). Among these shared mobility services, the commercial bike sharing service has grown to become a sustainable transportation alternative for millions across the globe, providing a low-carbon solution to the ‘last mile’ problem (Fishman et al. 2014; Nogal and Jiménez 2020; Wang et al. 2019). As Shaheen et al. (2019) suggested, the bike sharing service primarily comprises of three models: station-based, dock-less, and hybrid models. In a station-based model, users are required to return bikes to the bike stations, while in a dock-less model, users can pick up and return them within a predefined geographic region. A hybrid model integrates the characteristics of both the two models. In China, most bike sharing services have been adopting a mobile technology-based dock-less model that enables users to pick up and return bikes anywhere within an operating zone via mobile applications. A coin has two sides and so does the dock-less model; this free-floating bike sharing system enhances user flexibility and convenience, but at the same time, it gives rise to high operating costs with respect to re-allocating bikes for users on demand, monitoring users’ misbehaviors, fixing bikes, and so forth. Hello Bike is a firm that survives in a fierce competition in China (van Waes et al. 2020). In order to improve its business performance, Hello Bike has implemented a number of user-centered marketing activities, such as providing membership cards to users, employing customer relationship management programs, and encouraging users to report bike defects and/or other users’ violations. Moreover, big data analytics has made it possible to clearly inform users of proper parking zones for pick-ups and returns and minimize the operating cost by allowing users to do the tasks employees have to do.
Bike sharing, as one representative type of shared mobility, is essentially an access-based service that has transformed the understanding of possession/ownership and has reminded the importance of individual obligations that underlie shared objectives. Access-based services refer to services that allow users to access a resource for a defined period of time in return for an access payment while the service provider retains full legal ownership of the resource (Bardhi and Eckhardt 2012; Schaefers et al. 2016).
On page of 14, in the “5.3 Practical Implications” section:
Third, this study provides evidence that bike sharing firms should pay attention to building strong relationships with their customers. The loyal user is shown to be more likely to be involved in extra-role participation, implying that enhancing the user-provider relationship is a cost-efficient way to lead users to engage in citizenship behavior. Thus, firms should improve their data analysis and bike redistribution capabilities to promote bike accessibility, ensuring that bikes are always available (Song et al. 2020). As vehicle damage caused by normal wear and tear or user misbehavior is another constraint for bike availability, firms should promptly respond to user feedback and maximize bike availability. In short, commercial bike sharing firms should continuously strive to improve service quality (Macioszek et al. 2020). Bike sharing firms that effectively design customer loyalty program tactics to provide users with different kinds of rewards could also promote user satisfaction.
R3. 4. On the "Figure 1. Research Model" the legend should be added, because on the Figure are acronyms "H1a, H1b, H2a, H2b, H3a, H3b, etc." without explaining their meaning. In fact, these hypotheses are described in the paper text, but for readers, their meanings should also be explained on the figure.
AR3. 4. Thank you for raising this issue. To appropriately address this concern, we have decided to move the Figure 1 to the end of “2. Conceptual Background and Hypotheses Development” section, which allows readers to understand the meanings of acronyms (e.g., H1a, H1b, etc.) first and then have a look at Figure 1. By doing so, we believe that adding the legend in Figure 1 becomes unnecessary.
R3. 5. It would be better to not divide section 3 into further subsections, i.e. "3.1 Measurements" and "3.2. Data Collection", because their content is too short. In this case, it would be better to contain all in section 3 (without further divisions).
AR3. 5. Thank you for your constructive suggestion. We are also well aware that the subsections of 3.1 and 3.2 are relatively short. We however believe that the parts of measurements and data collection of this study should remain separate for a clear understanding. Moreover, currently, after this revision, we have longer subsections by adding some more discussions and relocating Table 2, which were in Appendix in an earlier version of manuscript. We hope that you can concur with this decision.
R3. 6. In the text of the article, the formal language should be used, using impersonal phrases, i.e. "done" (instead of we did), "prepared" (instead of we prepared), "worked out" (instead of we developed), etc. So, please correct these phrases throughout the article, e.g. line 302 ... "For our study" ..., line 319, ..."We utilized "..., etc.
AR3. 6. Thank you for your valuable comments. We do agree with you that formal languages should be used, avoiding personal phrases for an academic purpose. Thus, throughout the entire manuscript, all sentences with personal phrases have been rewritten, using impersonal phrases.
R3. 7. Information about survey research presented in section 3 should be presented in a more detailed way, line 313-315-Authors wrote that ..."313495 questionnaires were submitted of which 438 valid and complete samples were acquired. Many of questionnaires were not included further in the study because they were not valid and complete. Could Authors develop this description of rejected questionnaires? In addition, in the Discussion section, Authors wrote about the advantages of the "the snowball sampling method", but based on the collected data, it can be seen that a lot of surveys have been rejected. Authors should provide comments in this area.
AR3. 7. Thank you for your comments. The exact number of discarded questionnaires and the reasons of exclusion have been explained in more details in the “3.2 Data Collection” section. Moreover, the reasons for using ‘a snowball sampling technique’ have been further discussed in the revised version of this paper. We hope that our efforts are satisfactory.
On pages 9 and 10, “Data Collection” section:
This study used an online survey method for data collection. The units of analysis were commercial bike sharing service users in China. The online survey was conducted via Sojump (www.sojump.com), which is a popular Chinese survey website. A virtual snowball sampling technique was employed to share and forward the survey links on WeChat, a dominant Chinese social networking service, which can help collect data from a general population in China. According to Baltar and Brunet (2012), a virtual snowball sampling technique is suitable for data collection in the social media era, although it may cause some concerns for community biases, non-random sampling errors, or lacks of control. Further, in order to attract more attention from respondents, 1-2 CNY rewards were randomly provided to them. The period of data collection lasted about one month (from January 2018 to February 2018). Respondents were first requested to report the most frequently used bike sharing service as well as the duration of use on the opening page of the questionnaire. A total of 495 questionnaires were submitted of which 438 valid and complete ones were acquired. Out of the 495 questionnaires collected, 57 were excluded due to incomplete responses with missing service brand name or duration of use, or aberrant responses lacking justification. The respondents of the discarded questionnaires were not statistically different from those of the included questionnaires regarding the sample demographics. In addition, the service names of bike sharing such as Mobike and ofo, which explain about 90 percent (89.7%) of the sample in the study, were the top two biggest and representative bike sharing services in China during the period of data collection. Demographic information of the sample is shown in Table 3.
R3. 8. Appendix "Operational Definition and Measurement Items for Each Construct" should be included in section 3. Moreover, more descriptive information about this questionnaire should be provided.
AR3. 8. Thank you for your helpful suggestion. The table which were in Appendix has been relocated in the “3.1 Measurements” section as Table 2. In addition, more detailed descriptive information about the questionnaires have been added in this section. We hope that we have properly addressed this review point.
On page 8, in the “3.1 Measurements” section:
For this study, a survey instrument was designed to obtain data on research variables. There are eight constructs, of which the measurement items were drawn from previous studies and slightly modified to ensure the appropriateness for this study. All of the constructs were measured with multiple items based on a seven-point Likert scale (1 = strongly disagree, 7 = strongly agree). The operational definitions of all the constructs, the measurement items, and the related sources are presented in Table 2.
Newly Added References
Remarks: We appreciate the reviewer’s kind and sincere suggestions regarding the relevant and useful references like the followings. We have tried to add these resources to our revised paper. Thank you.
Macioszek, E.; Świerk, P.; Kurek, A. The Bike-sharing system as an element of enhancing sustainable mobility—A case study based on a city in Poland. Sustainability 2020, 12, 3285.
Nogal, M.; Jiménez, P. Attractiveness of bike-sharing stations from a multi-modal perspective: The role of objective and subjective features. Sustainability 2020, 12, 9062.
Shaheen, S.; Cohen, A.; Randolph, M.; Farrar, E.; Davis, R.; Nichols, A. Shared Mobility Policy Playbook. Retrieved from https://escholarship.org/uc/item/9678b4xs, 2019.
Song, M.; Wang, K.; Zhang, Y.; Li, M.; Qi, H.; Zhang, Y. Impact evaluation of bike-sharing on bicycling accessibility. Sustainability 2020, 12, 6124.
van Waes, A.; Farla, J.; Raven, R. Why do companies’ institutional strategies differ across cities? A cross-case analysis of bike sharing in Shanghai & Amsterdam. Environmental Innovation and Societal Transitions 2020, 36,151-163.

Round 2
Reviewer 2 Report
Authors have addressed reviewer comments.
Author Response
We would like to thank you for your prompt and positive reply to our revised paper. Following your note that English language and style are fine/minor spell check required, we carefully checked and tried to improve the English language and style, editing our manuscript thoroughly. We have highlighted the changes in the revised version by using the red-colored texts. Thank you.